# Game analysis on PPP model operation of abandoned mines ecological restoration under the innovation of central government's reward and punishment system in China

**Dongmei Feng[1], Liang Wang [1]\*, Xiumei Duan[2]**

1 School of Business Administration, Liaoning Technical University, Huludao, China, 2 College of Chemistry and Environment Engineering & Center of Instrument Analysis, YingKou Institute of Technology, Yingkou, China

\* 472010079@stu.lntu.edu.cn

**Data Availability Statement:** All relevant data are within the manuscript and its Supporting Information files.

## Abstract

There is a huge funding gap in the abandoned mines ecological restoration in China. It is of great research value to explore how PPP model can better introduce social capital into the low-profit ecological restoration of abandoned mines. Based on the innovation perspective of the central government's reward and punishment system, this paper constructs an evolutionary game model of "local government-social capital", analyzes the interaction and behavior mechanism of core stakeholders in the operation process of abandoned mines ecological restoration PPP mode, and discusses the influence of evolutionary equilibrium strategy and parameters change on evolutionary strategy under different scenarios by Matlab simulation. The research shows that the abandoned mines ecological restoration needs the support of the central government. When the local government lightly punishes the low-quality service of social capital, the central government needs to pay higher costs to promote all parties to actively participate in the operation and supervision of the PPP project. The revenue and cost of government supervision, the operating subsidy for social capital and the cost saved by social capital in providing bad service are the key factors affecting the evolution of the game between government and social capital. Punishment can effectively spur social capital to keep the contract and operate in the project, but the punishment effect will be ineffective without government supervision. Finally, some suggestions are put forward, such as establishing a long-term supervision mechanism and a reasonable income mechanism for PPP projects, increasing penalties for violations, attracting third parties to reduce supervision costs and strengthening communication between the two parties, so as to make the project take into account the economic performance of social capital and the social welfare of government departments, and achieve dual Pareto improvement.

## 1, Introduction

Mineral resources provide a strong material guarantee for China's economic development. However, the mining process of mineral resources will occupy a lot of land resources, and the

**Funding:** This research was funded by The key project of the Ministry of Science and Technology of China in 2018, "Environmental Geohazard Risk Assessment of West Open Pit Mining Area" (2017YFC1503102), Liaoning Provincial Education Department Social Science Project "Research on Early Warning and Control Measures of Ecological Security in Resource-Depleted Cities from the Perspective of Transformation – Taking Fushun City as an Example" (LJKR0141) and 2023 refers to Preliminary No. 1 Central Support for Local Reform-Humanities and Social Sciences 23-A032, "Study on the Evolution and Optimization of Ecological Networks in Fushun Urban Area under the Influence of Mining Ecological Management" (No. 55230010A032). The funders had no role in study design, data collection and analysis, decision to publish, or preparation of the manuscript.

**Competing interests:** The authors have declared that no competing interests exist.

abandoned mines and surrounding wasteland will lead to serious environmental pollution, geological disasters and ecological imbalance. Therefore, the abandoned mines ecological restoration is imperative [1]. After 2018, the central government of China will incorporate the construction of ecological civilization into the overall strategic layout of building a society with China characteristics, and the construction of ecological environment has been raised to a new height. Among them, the problem of mine and surrounding environment damage in resource-based cities needs to be solved, but the capital demand is huge and mainly borne by finance, and the restoration efficiency is not ideal. According to statistics, in the decade from 2005 to 2015, the central government, local governments and social enterprises invested 90 billion RMB in mine ecological restoration, and the restoration area exceeded 800,000 hectares. However, there are still more than 2.2 million hectares of mine wasteland that need to be restored urgently, which also means that trillions of funds need to be invested in mine ecological restoration projects.

Ecological restoration of abandoned mines involves utilizing external forces to facilitate the recovery, reconstruction and improvement of damaged ecosystems, aiming to return them to their natural state before mining or adapt them for certain beneficial human purposes. In China, the initiative for ecological restoration of abandoned mines began in the early 1980s. Currently, research literatures on this topic are predominantly concentrated on mining environmental ecological restoration technologies, methods, benefits of soil ecological restoration in mining areas and quality evaluation of restoration [2–9]. In recent years, attention has also been directed towards the goals and directions of abandoned mine site restoration, market-oriented operation of restoration projects, government regulations, measures and management methods in this process [10–12]. Due to the disorderly development and utilization of mines in the early stages in China, a considerable amount of environmental pollution has been generated without clear responsibility attribution. Presently, most mining environmental restoration and governance projects are government-led public welfare initiatives. However, they lack a sound profit mechanism, making it challenging to recover investments through public fees, resulting in significant financial conflicts in governance funding supply and demand. Faced with these challenges, local governments urgently need to explore new environmental governance models. The current national policy encourages private capital to invest in public domains such as abandoned mine environmental governance.

PPP (Public-Private Partnership) model is a long-term collaborative framework established between the government and social capital in the infrastructure and public services sectors, representing an innovative approach to government investment and financing. Through the PPP model, the government can engage social capital to effectively activate idle funds, alleviate fiscal expenditure pressures, and construct infrastructure and public welfare facilities that would traditionally be undertaken by government finances. In the context of environmental protection in China, the practical application of the PPP model has mainly focused on sectors with stable profit returns. Currently, ecological environmental restoration of many mines in China lacks a robust profit mechanism, and social capital is generally in a phase of observation and exploration [13, 14]. This situation is primarily manifested in the practical implementation of projects. On one hand, mine restoration projects typically involve substantial financial investments. Due to issues such as insufficient and asymmetric information in the public-private collaboration process, social capital may engage in rent-seeking behavior to secure higher investment returns, disregarding the public-private contract, or may compromise the quality of project operations to reduce costs. On the other hand, the absence of government involvement in the operation of mine restoration PPP projects, particularly the lack of a quality monitoring system after the public-private contract is established, results in these projects operating in a state of "blind supervision" and "lack of regulation." This leads to varying levels of service

quality in the operation of mine restoration PPP projects. These factors collectively impact the sustainable operation and development of mine restoration PPP projects. The question of how the government can enhance the quality of environmental governance services in mine areas by regulating private sector compliance with public-private contract agreements remains. Additionally, the influence of various factors under government supervision (such as operational subsidies, construction service costs, penalty severity, regulatory revenue and regulatory costs) on the strategic choices of government regulatory departments and social capital, and how to control these factors to guide both sectors towards an "ideal" state, require further exploration.

Considering that the central government plays an important role in financial and policy support for mine ecological restoration in resource cities, firstly, in this paper the PPP model innovation of the abandoned mines ecological restoration is aimed, the conflict of interests between local government(hereinafter referred to as government) and social capital is analyzed, and an evolutionary game model of mine ecological restoration in resource cities is constructed under the reward and punishment system of the central government as a tool in Section 3; Secondly, from the perspectives of government and social capital, the stability and evolution law of the cooperative evolution are studied between government and social capital driven by their own interests and constrained by external conditions in Section 4; Thirdly, the numerical simulation is carried out on the scenario evolution of government and social capital under different disturbance intensities of influencing factors to explore a new application of PPP model in the abandoned mines ecological restoration in Section 5; Finally, the sixth section provides an innovative and beneficial suggestion for the application of PPP project in mine ecological restoration.

## 2, Literature review

The PPP model has been widely applied in various public infrastructure construction sectors in China, such as urban transportation facilities and wastewater treatment, yielding positive outcomes [15, 16]. Scholars have explored multiple aspects of PPP, with previous research primarily focusing on risk identification, assessment and management control in PPP projects. For instance, Darrin et al. classified risks in PPP projects and evaluated them using Monte Carlo simulation [17]. Arndt et al. suggested a three-way risk-sharing approach among the government, enterprises and end consumers to enhance project execution [18]. Peter et al. conducted a systematic study and analysis of project risks related to social capital financing in the UK's Skye Bridge project [19]. Jonathan et al. identified risks in utility projects under the PPP model from the government's perspective and conducted risk analysis [20]. The PPP partnership involves multiple stakeholders, making the issue of benefit distribution one of the most prominent and challenging topics in PPP project collaboration [21]. Evolutionary Game Theory, an interdisciplinary theory combining evolutionary biology and rational economics, is suitable for in-depth research on the interactive mechanisms formed among participating entities due to benefit distribution issues. The field of Evolutionary Game Theory has seen increasing richness in research outcomes. Scholars, in addition to studying fundamental theories, have conducted interdisciplinary research by integrating practical applications. For example, Zhang et al. established an evolutionary game model for government and manufacturers under quantity control and trading, analyzing the impact of government policies on manufacturer decisions and the dynamic trends of quantity control and trading markets [22]. Shi et al. constructed an evolutionary game model based on social networks to explore the crucial influences of strategic choices by enterprises and decisions by consumers on the diffusion of low-carbon technologies [23]. Jiang et al. utilized a fractional-order game model to discuss the

unique, local and global stability of the evolutionary dynamics of green low-carbon innovation in manufacturing enterprises [24]. Currently, a hotspot of research involves the coordination and distribution of interests among PPP stakeholders. For instance, Medd et al. established a game model for risk sharing in PPP projects, analyzing how the government and enterprises can more reasonably share risks and proposing corresponding countermeasures [25]. Loosemore et al. outlined basic principles for risk sharing among parties in PPP projects and emphasized the game relationships between the government and enterprises in terms of cooperation mechanisms, core competencies and balanced interests, proposing a PPP project benefit distribution model [26] Hu et al. modified the PPP project benefit distribution model using the Shapley value method from the perspective of stakeholders' theory, aiming to maximize the interests of all parties through coordinated benefit distribution [27]. Wu et al. employed a Stackelberg game model to study how, under government compensation, PPP projects can achieve both investor-required returns and maximize social benefits [28]. Zhou et al. established a game model for PPP projects in land reclamation, analyzing the project's operational mechanism based on the cost-benefit function of agricultural land reclamation investments [29]. Yin et al. utilized Evolutionary Game Theory to analyze the evolutionary equilibrium strategies and pathways of various parties in responding to opportunistic behavior by contractors in public projects under short-term and long-term cooperation scenarios [30].

At the theoretical level, there is a relative lack of research on internal cooperation mechanisms, behavioral choices and strategic interactions among stakeholders in the application of the PPP model for ecological restoration of abandoned mine sites. Ongoing research in the field of PPP model application for mine site ecological restoration has yielded limited research outcomes. For instance, Yang et al. constructed a foundational analysis framework for the evolution game between the government and social capital in the PPP model for ecological restoration of abandoned mine sites, considering aspects such as the division of responsibilities, bidding processes, risk-sharing, benefit sharing, and regulatory perspectives [31]. Liu et al. analyzed the possibilities and challenges of PPP application in mine site ecological restoration, conducted a SWOT analysis, and explored specific models and profit mechanisms for PPP application [32, 33]. However, the literatures mentioned above primarily explore internal game mechanisms and behavioral choices within the existing PPP framework. In practice, not only do local governments guide the implementation of abandoned mine site restoration projects, but the central government also plays a crucial role. It is of great research value to explore how the PPP model, based on the central government's system of rewards and penalties, can better integrate social capital into low-profit ecological restoration projects for abandoned mines. Investigating the application of innovative PPP models in low-profit mine site ecological restoration projects has significant research value. Exploring how PPP innovation models can be effectively applied in the ecological restoration of low-profit mine sites under the perspective of central government reward and penalty systems represents an important avenue for research.

This study employs mathematical analysis to investigate the interactive and behavioral mechanism evolution in the operation of the PPP models for ecological restoration of abandoned mine sites. The research features the following two aspects:

1. Based on the limited rationality of game participants, the study treats the operational process of PPP projects for ecological restoration of abandoned mine sites as a dynamic evolutionary process. It introduces the "reward-penalty" mechanisms of the central government in PPP project operation supervision as variables in the model along with various factors under government regulation. The study constructs an evolutionary game model between government regulatory departments and social capital, analyzing the impact and influence

of various factors on the strategic choices of government regulatory departments and social capital.

2. The research delves into the game relationship and strategic adjustments between government regulatory departments and social capital. Using Matlab for simulation analysis, the study examines the evolutionary equilibrium strategies and the influence of model parameter adjustments on the behavioral strategy choices of both parties in different scenarios. This approach provides a more intuitive understanding of the dynamic evolutionary trends in the operation of PPP models for ecological restoration of abandoned mines. Based on the conclusions drawn from the analysis, practical measures to enhance the quality of PPP project operation are suggested from the perspective of government regulation.

## 3, Model hypothesis and construction

For the convenience of model construction and subsequent analysis, this study first makes some basic hypotheses based on the analysis of previous PPP model operations:

Upon studying the characteristics of government PPP project operations, it is observed that in the PPP projects for ecological restoration of abandoned mines, the BOT (Build-Operate-Transfer) and O&M (Operation & Maintenance) approaches are utilized to provide services throughout the project lifecycle [34]. To establish a safeguard system for the ecological restoration of mines under the PPP model and create a fair market order, it is assumed that the government has entrusted a third-party professional consulting agency to establish a standardized quality evaluation system for the governance, restoration and operation required by the mines ecological restoration PPP project. The government assigns a specialized supervisory department, such as the Natural Resources Department or the Industrial Park Management Committee, to oversee the implementation and operation of the PPP project. Additionally, the government delegates the task of periodically assessing the service quality provided by social capital to a third-party consulting agency to ensure the maximization of social welfare for the government. Social capital is responsible for the ecological restoration of the mines and subsequent project operations. It obtains returns through operational subsidies, user payments, or multi industry implantation. Therefore, the following hypothesis is made:

**Hypothesis 1**: The core stakeholders in the game include government departments and social capital. We assume no differentiation among them as players in the game. In the presence of information asymmetry, both parties are considered limited rationality groups, where each lacks complete understanding of the other's strategies and payoff functions. Throughout the game, stakeholders continuously adjust their strategies until they find the optimal ones. While government departments seek to maximize overall social welfare, they also pursue their own interests. On the other hand, social capital aims to maximize profits and shareholder equity.

Analysis reveals two behavioral strategies adopted by social capital in PPP projects for abandoned mines restoration. First, ensuring the high-quality and efficient execution of the entire PPP project from construction to operation, meeting the financial pressures and social welfare needs of local governments. Second, driven by profit motives, social capital may engage in speculative activities such as violations, rent-seeking and falsification during project construction and commissioned operations, which may harm the project or public interests. In abandoned mines restoration PPP projects, the government needs to implement effective incentives, constraints and supervision throughout the entire project life cycle (from construction to commissioned operations) to encourage positive collaboration by social capital, and

supervise speculative behaviors. Due to objective factors such as the lack of specialized management capabilities in PPP projects and information asymmetry, the government may face high supervision costs, leading to a tendency toward non-supervision behavior. Therefore, the following assumption is made:

**Hypothesis 2**: It is assumed that the strategy space for social capital is {good service, bad service}, with probabilities of selecting "good service" and "bad service" denoted as x and 1-x, respectively. The assumption is made that the local government's strategy space is {supervision, non-supervision}, with probabilities of adopting the "supervision" and "non-supervision" strategies denoted as y and 1-y, respectively.

**Hypothesis 3:** According to the commitment clauses of the company contract signed by the PPP project, it is assumed that the realizable fixed income of social capital is reported as $V_e$. If social capital chooses the strategy of "good service", the cost it needs to pay is $C_h$, and the special operating subsidy provided by the government can be obtained after the service quality is evaluated by the government supervision department or the entrusted third-party institution. If social capital chooses the strategy of "bad service", it needs to pay the cost $C_d$, at this time, $C_h > C_d$, but when the government supervision department executes supervision, it will be punished (fines or discounts required by the government), and the resulting losses will be $F_e$.

**Hypothesis 4:** Assume that the government supervision department can gain $V_g$ by choosing the "supervision" strategy, but it needs to pay the supervision cost $C_g$. If the government supervision department chooses "non-supervision", it is assumed that there will be non-supervision cost, but the government will not get the corresponding supervision income. However, when the social capital chooses "good service", the government needs to pay the social capital as $L_e$. If the social capital chooses to operate in breach of contract and provide bad services, and the government fails to supervise the violations due to inaction at this time, it will be reported by the public and punished $F_g$ by the central government to the local government.

When government guides social capital into mine ecological restoration projects and effectively supervise them, and social capital chooses to provide good service, that is, when all participants in mine ecological restoration can actively participate, the central government will reward all participants. In other cases, the central government will not reward any party. Therefore, the following assumption is made:

**Hypothesis 5:** Assume that the central government's rewards to local governments and social capital are $J_g$ (such as subsidies, bonuses and official position promotion and $J_g < F_e$) and $J_e$ (reputation, word of mouth and other spiritual rewards etc.).

Assuming that the variables in the above assumptions are all greater than 0, according to these hypothesis, the evolutionary game payment matrix formed between social capital and government supervision departments is shown in Table 1.

## 4, Model analysis

### 4.1, Equilibrium points of evolution process

In this paper, Friedman analysis method is introduced to construct the game model of PPP mode for abandoned mines ecological restoration, the dynamics evolution equations of replication are established and made stability analysis [35].

**Table 1. Payoff matrix for both players.**

| Social Capital | Government | |
|---|---|---|
| | Supervision ($y$) | Non-supervision ($1-y$) |
| Good service ($x$) | $V_e - C_h + L_g + J_e$, $V_g - C_g - L_e + J_g$ | $V_e - C_h + L_g$, $-L_e$ |
| Bad service ($1-x$) | $V_e - C_d - F_e$, $V_g - C_g + F_e$ | $V_e - C_d$, $-F_g$ |

According to the above evolutionary game payoff matrix, the expected values $u_{11}$ and $u_{12}$ of social capital's strategy "good service" and "bad service", and the average expected value $u_1$ of social capital are respectively:

$$\begin{cases} u_{11} = y*(V_e - C_h + L_g + J_e) + (1-y)*(V_e - C_h + L_g) = L_g - C_h + V_e + J_e*y \\ u_{12} = y*(V_e - C_d - F_e) + (1-y)*(V_e - C_d) = V_e - C_d - F_e*y \\ u_1 = x*u_{11} + (1-x)*u_{12} \end{cases} \tag{1}$$

By analyzing the replicator dynamics of social capital, the replicator dynamic equation of social capital is obtained:

$$F_x = \frac{dx}{dt} = x(u_{11} - u_1) = x(1-x)\left[(F_e + J_e)*y + C_d - C_h + L_g\right] \tag{2}$$

In the same way, the expected values $u_{21}$ and $u_{22}$ of the government strategies "supervision" and "non-supervision", and the average expected value $u_2$ of the government are as follows:

$$\begin{cases} u_{21} = x(V_g - C_g - L_e + J_g) + (1-x)(V_g - C_g + F_e) \\ u_{22} = x(-L_e) + (1-x)(-F_g) \\ u_2 = y*u_{21} + (1-y)*u_{22} \end{cases} \tag{3}$$

By analyzing the replicator dynamics of the government, the replicator dynamic equation of the government is obtained:

$$F_y = \frac{dy}{dt} = y*(u_{21} - u_2) = y(1-y)\left[(1-x)(F_g + F_e) + xJ_g - C_g + V_g\right] \tag{4}$$

According to the above analysis, a two-dimensional dynamic system $I$ can be obtained from the formulas $F_x$ and $F_y$, namely

$$I.\begin{cases} F_x = \frac{dx}{dt} = x(1-x)\left[(F_e + J_e)*y + C_d - C_h + L_g\right] \\ F_y = \frac{dy}{dt} = y(1-y)\left[(1-x)(F_g + F_e) + xJ_g - C_g + V_g\right] \end{cases} \tag{5}$$

In order to analyze the equilibrium points and stability of the system $I$ conveniently, we set:

$$x^* = \frac{F_g + F_e + V_g - C_g}{F_g + F_e - J_g}, \quad y^* = \frac{C_h - C_d - L_g}{F_e + J_e} \tag{6}$$

**Proposition 1:** The equilibrium points of this system are $(0, 0)$, $(0, 1)$, $(1, 0)$, $(1, 1)$, $(x^*, y^*)$.

**Proof:** In the above two-dimensional dynamic system $I$, we make $dx/dt = 0$, $dy/dt = 0$, Equilibrium points $(0, 0)$, $(0, 1)$, $(1, 0)$, $(1, 1)$ of the system $I$ are got. Substituting $(x^*, y^*)$ into system $I$, we can also get $dx/dt = 0$, $dy/dt = 0$. Therefore, system $I$ has five equilibrium points.

## 4.2, Analysis of equilibrium points and evolutionary stability strategy

According to Friedman's theory, Jacobian matrix $J$ of two-dimensional dynamic system $I$ is obtained:

$$J = \begin{bmatrix} A_{11} & A_{12} \\ A_{21} & A_{22} \end{bmatrix} \qquad (7)$$

Where $A_{11}$, $A_{12}$, $A_{21}$ and $A_{22}$ are respectively:

$$\begin{cases} A_{11} = \dfrac{\partial F_x}{\partial x} = (1 - 2x)\Big[(F_e + J_e)y + C_d - C_h + L_g\Big] \\[2mm] A_{12} = \dfrac{\partial F_x}{\partial y} = x(1 - x)(F_e + J_e) \\[2mm] A_{21} = \dfrac{\partial F_y}{\partial x} = -y(1 - y)\Big(F_g + F_e + J_g\Big) \\[2mm] A_{22} = \dfrac{\partial F_y}{\partial y} = (1 - 2y)\Big[(1 - x)(F_g + F_e) + xJ_g - C_g + V_g\Big] \end{cases} \qquad (8)$$

If the following two conditions can be met at the same time, the equilibrium points of replicator dynamic equation is the Evolutionary Stability Strategy (ESS):

$$TrJ = A_{11} + A_{22} < 0 \text{ (Trace condition)} \qquad (9)$$

$$DetJ = A_{11}A_{12} - A_{12}A_{21} > 0 \text{ (Determinant condition)} \qquad (10)$$

From Jacobian matrix, the values of $A_{11}$, $A_{12}$, $A_{21}$ and $A_{22}$ at five equilibrium points can be obtained, as shown in Table 2:

The local equilibrium points obtained by solving the replicator dynamic equation is not necessarily the ESS points. According to the values in Table 2, for the local equilibrium point $(x^*, y^*)$, it is obvious that $A_{11} + A_{22} = 0$ can be obtained which does not meet the trace condition of Formula (9), and it can't meet both conditions (9) and (10), therefore, this point is not the ESS point of the system $I$. Thus, we only need to analyze the determinant and trace at local equilibrium points $(0, 0)$, $(0, 1)$, $(1, 0)$ and $(1, 1)$, and accordingly analyze the stability of system $I$.

Table 2. **Specific values of $A_{11}$, $A_{12}$, $A_{21}$ and $A_{22}$ at local equilibrium points.**

| Equilibrium point | $A_{11}$ | $A_{12}$ | $A_{21}$ | $A_{22}$ |
|---|---|---|---|---|
| $(0, 0)$ | $C_d - C_h + L_g$ | $0$ | $0$ | $V_g - C_g + F_g + F_e$ |
| $(1, 0)$ | $-(C_d - C_h + L_g)$ | $0$ | $0$ | $V_g - C_g + J_g$ |
| $(0, 1)$ | $F_e + J_e + C_d - C_h + L_g$ | $0$ | $0$ | $-(V_g - C_g + F_g + F_e)$ |
| $(1, 1)$ | $C_h - C_d - F_e - J_e - L_g$ | $0$ | $0$ | $-(V_g - C_g + J_g)$ |
| $(x^*, y^*)$ | $0$ | $M^*$ | $N^*$ | $0$ |

$^*M$ and $N$ are two numbers that are not zero.

Note: $M = \dfrac{(F_g + F_e + V_g - C_g)(C_g - V_g - J_g)(F_e + J_e)}{(F_g + F_e - J_g)^2}$, $N = \dfrac{(C_h - C_d - L_g)(C_h - C_d - L_g - F_e - J_e)(F_g + F_e + J_g)}{(F_e + J_e)^2}$.

**Proposition 2**, which is divided into five situations:

Scenario1: when $L_g < C_h - C_d < F_e + L_g + J_e$ and $V_g + F_e + J_g < C_g - F_g$, or when $F_e + L_g + J_e < C_h - C_d$ and $V_g + F_e + J_g < C_g - F_g$, the ESS point of system $I$ is (0, 0).

Scenario 2: when $L_g < C_h - C_d < F_e + L_g + J_e$ and $C_g - F_g < V_g + J_g < C_g$, there is no ESS in system $I$.

Scenario 3: when $L_g > C_h - C_d$ and $V_g + F_e + J_g < C_g - F_g$, or when $C_h - C_d < L_g$ and $C_g - F_g < V_g + J_g < C_g$, the ESS point of system $I$ is (1, 0).

Scenario 4: when $F_e + L_g + J_e < C_h - C_d$ and $C_g - F_g < V_g + J_g < C_g$, or when $F_e + L_g + J_e < C_h - C_d$ and $V_g + J_g > C_g$, the ESS point of system $I$ is (0, 1).

Scenario 5: when $L_g + F_e + J_e > C_h - C_d$ and $V_g + J_g > C_g$, or when $L_g < C_h - C_d < F_e + L_g + J_e$ and $V_g + J_g > C_g$, the ESS point of system $I$ is (1, 1).

**Proof:** From the equations of two-dimensional dynamic system $I$, the values of determinant det$J$ and trace tr$J$ of the four equilibrium points (0, 0), (0, 1), (1, 0) and (1, 1) of Jacobian matrix can be obtained firstly, and then the local stability can be judged. The analysis of equilibrium points for five scenarios in Proposition 2 is shown in Table 3, and the detailed proof process can be found in the S1 Appendix.

## 5, Numerical simulation analysis

In order to analyze the operation mechanism of abandoned mine ecological restoration PPP mode more scientifically and intuitively, MATLAB is used to simulate all the scenarios analyzed in section 3. The parameters involved in the following sections need to meet the conditions mentioned above, and the PPP operation mode in different scenarios can be assigned according to different situations of the PPP project. The value does not represent the actual cost or income of government and social capital in the PPP project of mine ecological restoration.

### 5.1, Scenario verification of ESS

**(1) Scenario 1:**

Let the parameters be $Fe = 30$, $Fg = 10$, $Cg = 60$, $Lg = 30$, $Ch = 115$, $Cd = 60$, $Vg = 10$, $J_e = 5$, $J_g = 5$, which satisfies the condition of $L_g < C_h - C_d < F_e + L_g + J_e$ and $V_g + F_e + J_g < C_g - F_g$, or let the parameters be $Fe = 1$, $Fg = 10$, $Cg = 60$, $Lg = 30$, $Ch = 115$, $Cd = 60$, $Vg = 10$, $J_e = 5$, $J_g = 5$, which satisfies the condition of $F_e + L_g + J_e < C_h - C_d$ and $V_g + F_e + J_g < C_g - F_g$ in Scenario 1, and the results obtained by Matlab simulation are shown in Fig 1. Fig 1 shows that when $x$ and $y$ are in different initial proportions, a large proportion of social capital tends to choose the strategy of "good service" at first, but because the operating subsidy obtained is less than the speculative cost saved when choosing the strategy of "bad service" and the government's punishment for social capital "bad service" is light, it resulted in the net income of social capital choosing the strategy "good service" is less than the net income of choosing the strategy "bad service", Finally, which drives social capital from providing "good service" to "bad service" under the motivation of maximizing the interests and shareholders. At first, the government chose the strategy of "supervision" in order to maximize its own social interests. However, at this time, because the supervision cost of the government is higher than the supervision income, and the central government's punishment for local government's supervision dereliction of duty is light, this will drive the local government from "supervision" to "non-supervision", and the final result is that the interaction between the two parties has evolved to the worst equilibrium point (0, 0). In this scenario, the government

**Table 3. Equilibrium point analysis of five scenarios.**

| No. | Conditions | Equilibrium Point | (0, 0) | (1, 0) | (0, 1) | (1, 1) |
|---|---|---|---|---|---|---|
| S1 | $L_g < C_h - C_d < F_e + L_g + J_e$ & $V_g + F_e + J_g < C_g - F_g$ | trJ | - | ± | + | ± |
| | | detJ | + | - | + | - |
| | | ESS | ESS | Saddle | Unstable | Saddle |
| | $F_e + L_g + J_e < C_h - C_d$ & $V_g + F_e + J_g < C_g - F_g$ | trJ | - | ± | ± | + |
| | | detJ | + | - | - | + |
| | | ESS | ESS | Saddle | Saddle Point | Saddle |
| S2 | $L_g < C_h - C_d < F_e + L_g + J_e$ & $C_g - F_g < V_g + J_g < C_g$ | trJ | ± | ± | ± | ± |
| | | detJ | - | - | - | - |
| | | ESS | Saddle | Saddle | Saddle | Saddle |
| S3 | $L_g > C_h - C_d$ & $V_g + J_e + F_e < C_g - F_g$ | trJ | ± | - | + | ± |
| | | detJ | - | + | + | - |
| | | ESS | Saddle | ESS | Unstable | Saddle |
| | $C_h - C_d < L_g$ & $C_g - F_g < V_g + J_g < C_g$ | trJ | + | - | ± | ± |
| | | detJ | + | + | - | - |
| | | ESS | Unstable | ESS | Saddle | Saddle |
| S4 | $F_e + L_g + J_e < C_h - C_d$ & $C_g - F_g < V_g + J_g < C_g$ | trJ | ± | ± | - | + |
| | | detJ | - | - | + | + |
| | | ESS | Saddle | Saddle | ESS | Unstable |
| | $F_e + L_g + J_e < C_h - C_d$ & $V_g + J_g > C_g$ | trJ | ± | + | - | ± |
| | | detJ | - | + | + | - |
| | | ESS | Saddle | Unstable | ESS | Saddle |
| S5 | $L_g > C_h - C_d$ & $V_g + J_g > C_g$ | trJ | + | ± | ± | - |
| | | detJ | + | - | - | + |
| | | ESS | Unstable | Saddle | Saddle | ESS |
| | $L_g < C_h - C_d < F_e + L_g + J_e$ & $V_g + J_g > C_g$ | trJ | ± | + | ± | - |
| | | detJ | - | + | - | + |
| | | ESS | Saddle | Unstable | Saddle | ESS |

supervision mechanism is virtually non-existent, as the central government's commitment to reward local governments and social capital is too low and has not been able to reverse the situation. The service quality of abandoned mine ecological restoration PPP projects is at a low level, and the overall social utility is at its lowest level of "ineffectiveness". If this ineffective situation continues to develop, it will eventually lead to the failure of the sustainable construction and operation of the mines ecological restoration PPP project, and the project will end in failure.

**(2) Scenario 2:**

Let the parameters be $F_e = 20$, $Fg = 30$, $Cg = 50$, $L_g = 30$, $C_h = 120$, $C_d = 60$, $V_g = 20$, $Je = 20$, $Jg = 5$, which satisfies the condition of $L_g < C_h - C_d < F_e + L_g + J_e$ and $C_g - F_g < V_g + F_e < C_g$ in Scenario 2, the results obtained by Matlab simulation are shown in Fig 2(A). Fig 2(A) shows that when $x$ and $y$ are in different initial proportions, according to the steps of evolutionary iteration, if the cost saved by social capital when choosing the strategy "bad service" is greater than the sum of the operating subsidy and the penalty loss for violation when choosing the strategy "good service", social capital will choose the strategy "good service" when it is supervised by the government, but choose the strategy "bad service" when it is not supervised. However, when the supervision income of the government is greater than the difference between the supervision cost and the penalty loss of dereliction of duty, the behavior evolution

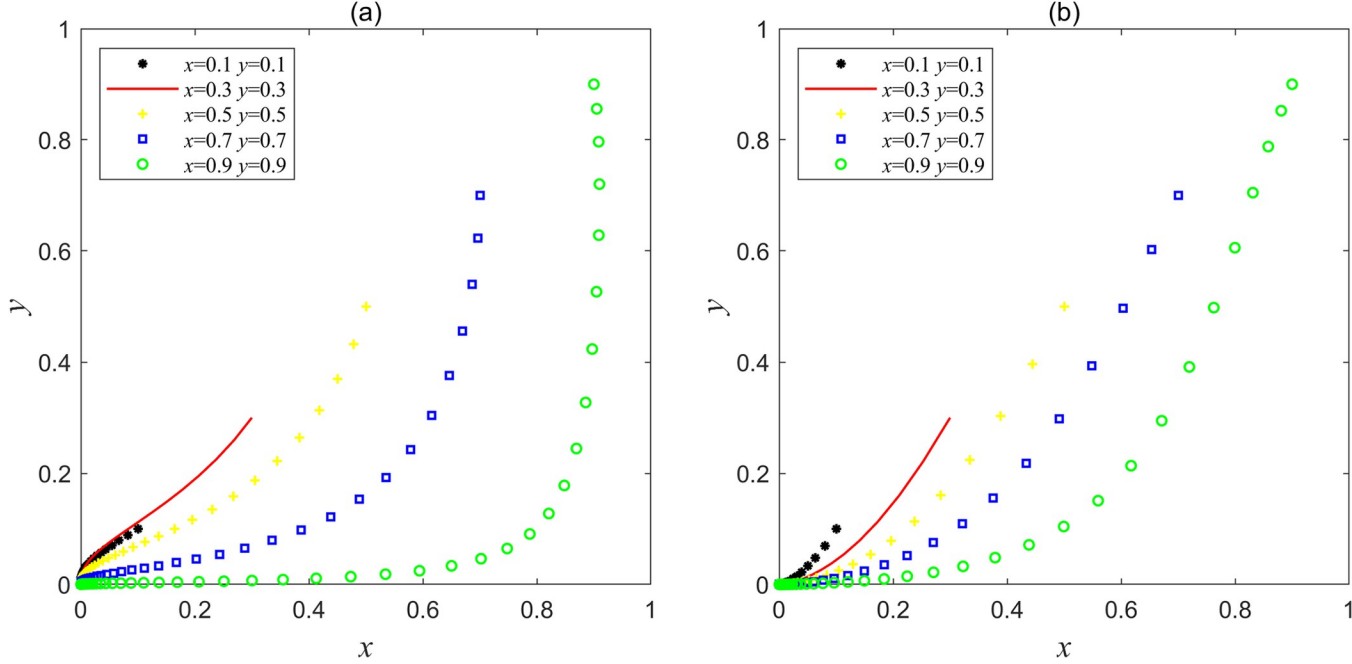

**Fig 1. Simulation results of stable point (0, 0) evolution in Scenario 1.**

trend of the government is similar with and the social capital, all the same, they are in a state of periodic swing and will not form a stable evolution strategy. As far as government departments are concerned, scholars have identified the failure mechanisms through numerous PPP project cases, pointing out that the lack of a sound and effective supervision system is one of the

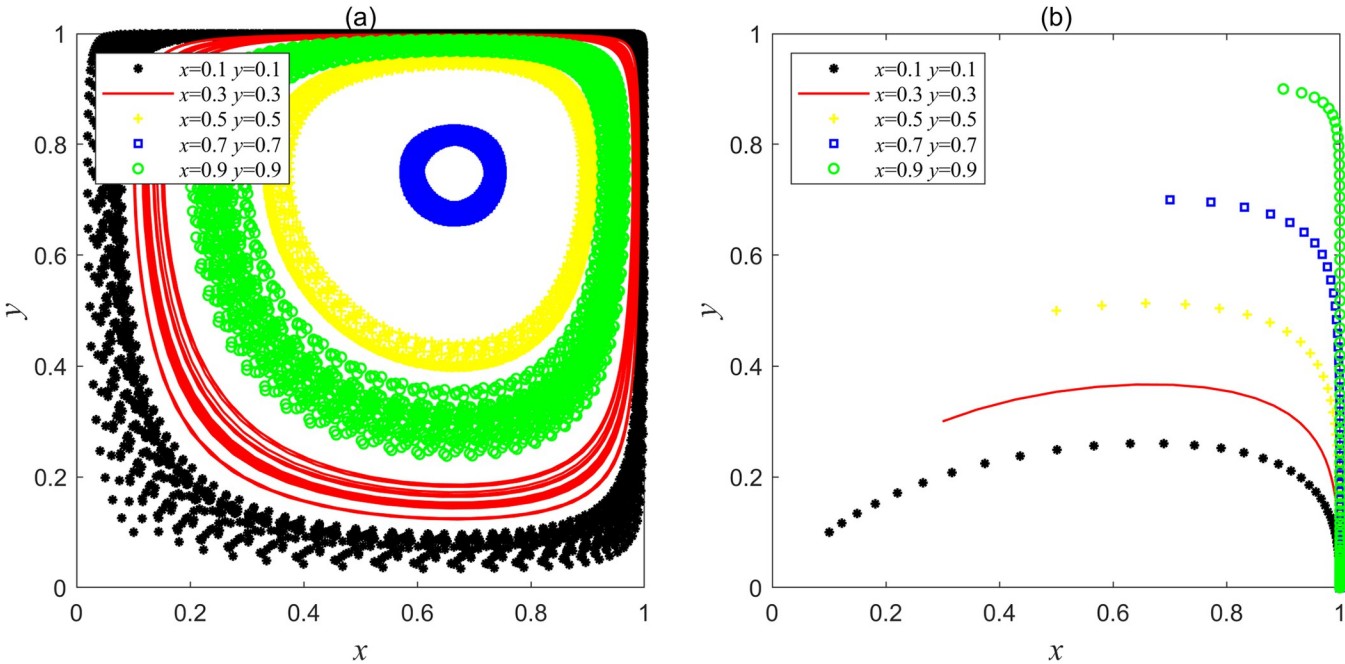

**Fig 2.** (a) Simulation results of no stable point evolution in Scenario 2; (b) Simulation results of evolution of stable point (1, 0) in Scenario 3.

important reasons for the failure [36, 37]. The simulation results of Scenario 2 explain that in reality, government departments adopt positive movement style supervision mode in pursuit of maximizing social interests in the high-incidence quality problems period of PPP project, while government departments adopt negative intermittent supervision mode in the low-incidence period in order to ensure the interests of their own. The central government's reward commitment to local governments and social capital can't make both sides stabilize their strategic choices in this scenario, and the government's regulatory mechanism can only temporarily curb the service quality issues of these PPP projects, but cannot promote social capital to improve the service quality of PPP projects in the long term and completely solve the service quality problems of PPP projects.

**(3) Scenario 3:**

Let the parameters be $F_e$ = 20, $Fg$ = 30, $Cg$ = 50, $L_g$ = 60, $C_h$ = 100, $C_d$ = 60, $V_g$ = 20, $J_e$ = 5, $J_g$ = 20, which satisfies the condition of $L_g > C_h - C_d$ and $C_g - F_g < V_g + J_g < C_g$ in Scenario 3, the results obtained by Matlab simulation are shown in Fig 2(B). Fig 2(B) shows that when $x$ and $y$ are in different initial proportions, a large proportion of social capital will choose the strategy "bad service" at the beginning, although the government's punishment for social capital's violations is relatively small, the operating subsidy of social capital that chooses the strategy "good service" is greater than the cost reduced by choosing the strategy "bad service", so the net income of choosing the strategy "good service" is greater than the net income of choosing the strategy "bad service", and finally social capital tends to "good service" driven by maximizing its own and shareholders' interests. In order to maximize the social benefits, a large proportion of governments will choose the strategy "supervision" at first, but at this time, because the difference between the supervision cost of government and the rewards received by the central government is greater than the income from supervision, it will drive government to choose to trust social capital in consideration of their own interests, from "supervision" to "non-supervision", and the central government's rewards to local governments have not played a corresponding incentive role. The interaction between the two parties has evolved to the sub-optimal equilibrium point (1, 0). At this time, the government supervision mechanism is in a bad "locked" state. If the government fails to supervise for a long time, the high operating subsidy may become a means of "enclosure of money" for someone (including social capital and government officials), thus, the central government's special governance funds will be "cashed out" and the actual mine governance cost will rise very high, which is a huge waste of the overall national funds.

**(4) Scenario 4:**

Let the parameters be $F_e$ = 10, $Fg$ = 25, $Cg$ = 40, $L_g$ = 10, $C_h$ = 120, $C_d$ = 60, $V_g$ = 15, $J_e$ = 20, $J_g$ = 1, which satisfies the condition of $F_e + L_g + J_e < C_h - C_d$ and $C_g - F_g < V_g + J_g < C_g$, or let the parameters be $F_e$ = 20, $Fg$ = 30, $Cg$ = 10, $L_g$ = 10, $C_h$ = 120, $C_d$ = 60, $V_g$ = 20, $J_e$ = 20, $J_g$ = 10, which satisfies the condition of $F_e + L_g + J_e < C_h - C_d$ and $V_g + J_g > C_g$ in Scenario 4, the results obtained by Matlab simulation are shown in Fig 3. Fig 3 shows that when $x$ and $y$ are in different initial proportions, a higher proportion of social capital will choose the strategy of "good service" at first, but the sum of the operating subsidy, the promised reward and punishment of the central government is less than the cost of speculation reduction brought by choosing the strategy "bad service", and the government's punishment for "bad service" of social capital is lighter. As a result, the net profit of choosing the strategy "good service" is less than that choosing the strategy "bad service", which ultimately drives social capital to turn from "good service" to "bad service" considering the maximization and shareholders' interests. At first, the government showed the behavior of trusting social capital, and a high proportion chose the strategy "non-supervision". However, with the proportion increase of the social

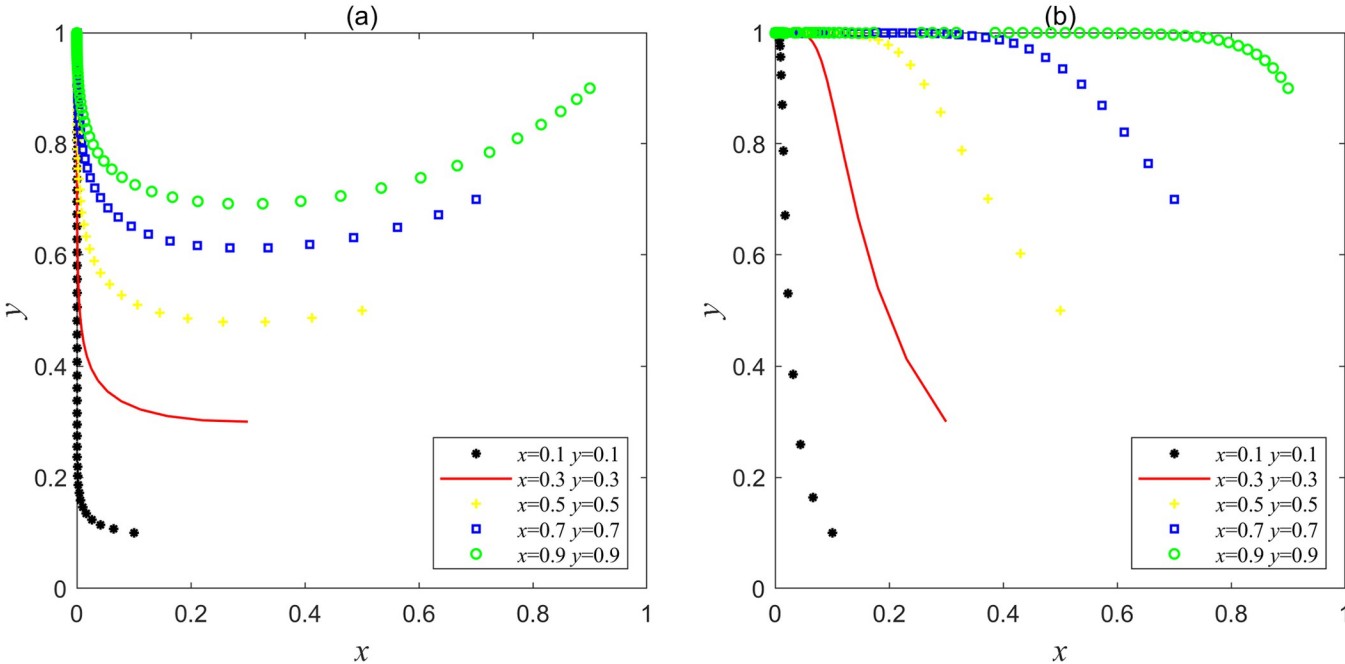

**Fig 3. Simulation results of evolution of stable point (0, 1) in Scenario 4.**

capital choosing the strategy "bad service", the reduction of supervision cost will make the supervision net income of the government lower than that of the "non-supervision". At this time, the government will pursue its own interests while pursuing the maximization of social interests, and finally move from "non-supervision" to "supervision". This scenario will make the interaction between the two parties evolve to the second-worst equilibrium point (0, 1). At this time, the government supervision mechanism will also be in a bad "locked" state. Although the government will actively supervise, it will only curb the illegal behavior of social capital in a short time, and will not fundamentally promote social capital to improve the quality of mine ecological restoration services, which eventually affect the high-quality development of mine ecological restoration PPP project.

**(5) Scenario 5:**

Let the parameters be $F_e = 20$, $Fg = 30$, $Cg = 40$, $L_g = 50$, $C_h = 100$, $C_d = 60$, $V_g = 30$, $J_e = 5$, $J_g = 20$, which satisfies the condition of $L_g > C_h - C_d$ and $V_g + J_g > C_g$, or let the parameters be $Fe = 20$, $Fg = 30$, $Cg = 40$, $L_g = 30$, $C_h = 120$, $C_d = 60$, $V_g = 30$, $J_e = 20$, $J_g = 20$, which satisfies the condition of $L_g < C_h - C_d < L_g + J_e + F_e$ and $V_g + J_g > C_g$ in Scenario 4, the results obtained by Matlab simulation are shown in Fig 4. Fig 4 shows that when $x$ and $y$ are in different initial proportions, a higher proportion of social capital will choose the strategy "bad service" at first, because the sum of the operating subsidies, the promised rewards and penalties of the central government is higher than the cost of speculative reduction by choosing the strategy "bad service", or the government's punishment for violations of social capital is increased, so the net income of choosing the strategy "good service" will be higher than that of choosing "bad service". However, when the government initially trusts social capital behavior, a higher proportion will choose the strategy of "non-supervision". With the reduction of the supervision cost of the government, the net income of supervision" is less than that of "non-supervision", so in the end, the government pursues its own interests while pursuing the maximization of social interests, and tends to "supervision" from "non-supervision". This shows that the central

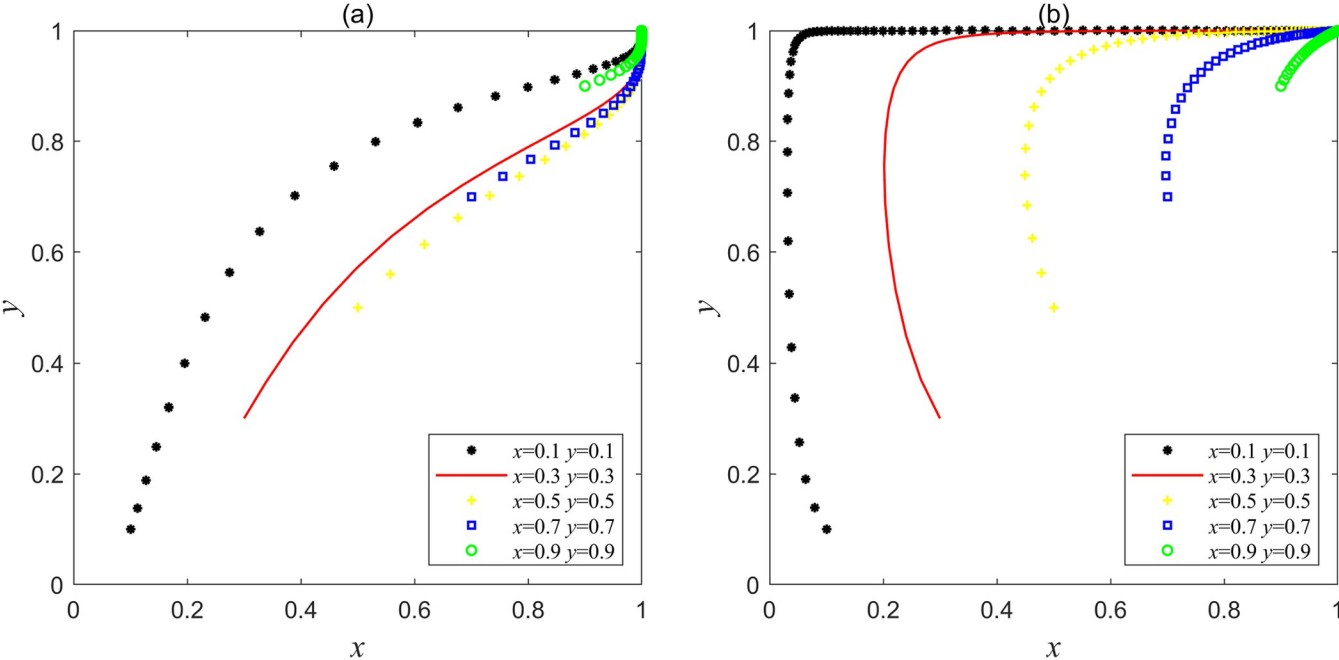

**Fig 4. Simulation results of evolution of stable point (1, 1) in Scenario 5.**

government's incentive commitment, operating subsidy mechanism and punishment mechanism for social capital are in effect. With the reduction of regulatory costs, the increase of regulatory revenue and the incentive commitment for the central government, the government is actively supervising, and the interaction between the two parties has evolved to the optimal equilibrium point (1, 1). In this scenario, the government supervision mechanism has reached an "ideal" state, and the social capital and the enthusiasm of government are relatively high. One party can consciously and actively focus on improving the quality of mine ecological restoration services, the other can give full play to the advantages of supervision resources, maximize social benefits, which achieve sustainable and high-quality development of mine ecological restoration PPP project.

## 5.2, Simulation analysis of model variable adjustment

In order to better explore how the government effectively supervises social capital, it can finally push the evolution of the game to the ideal mode, that is, both players of the game can finally evolve to the optimal equilibrium point (1, 1) of social capital choosing strategy "good service" while the government chooses the "supervision" strategy. By adjusting the variable values of the established model in Section 4.2, we will see how to promote the evolution results of the three "non-ideal" states of strategy Scenario 1, Scenario 3 and Scenario 4 discussed in Section 4.2 to evolve to the target state of Scenario 5, and analyze the influence of the various parameters changes on the evolution results. It is assumed that the initial values of $x$ and $y$ are both 0.3.

**(1) Variable adjustment in Scenario 1.** Based on the value of the parameter variables in Scenario 1, let $(L_g, C_g)$, $(L_g, V_g)$, $(C_d, V_g)$ and $(F_e, F_g)$ be four groups of variables, and carry out Matlab simulation calculation on these four groups of variables respectively. Results are shown in Fig 5, with the increase of operating subsidies for social capital to choose the strategy "good service" or the increase of speculative costs for choosing the strategy "bad service", at the same

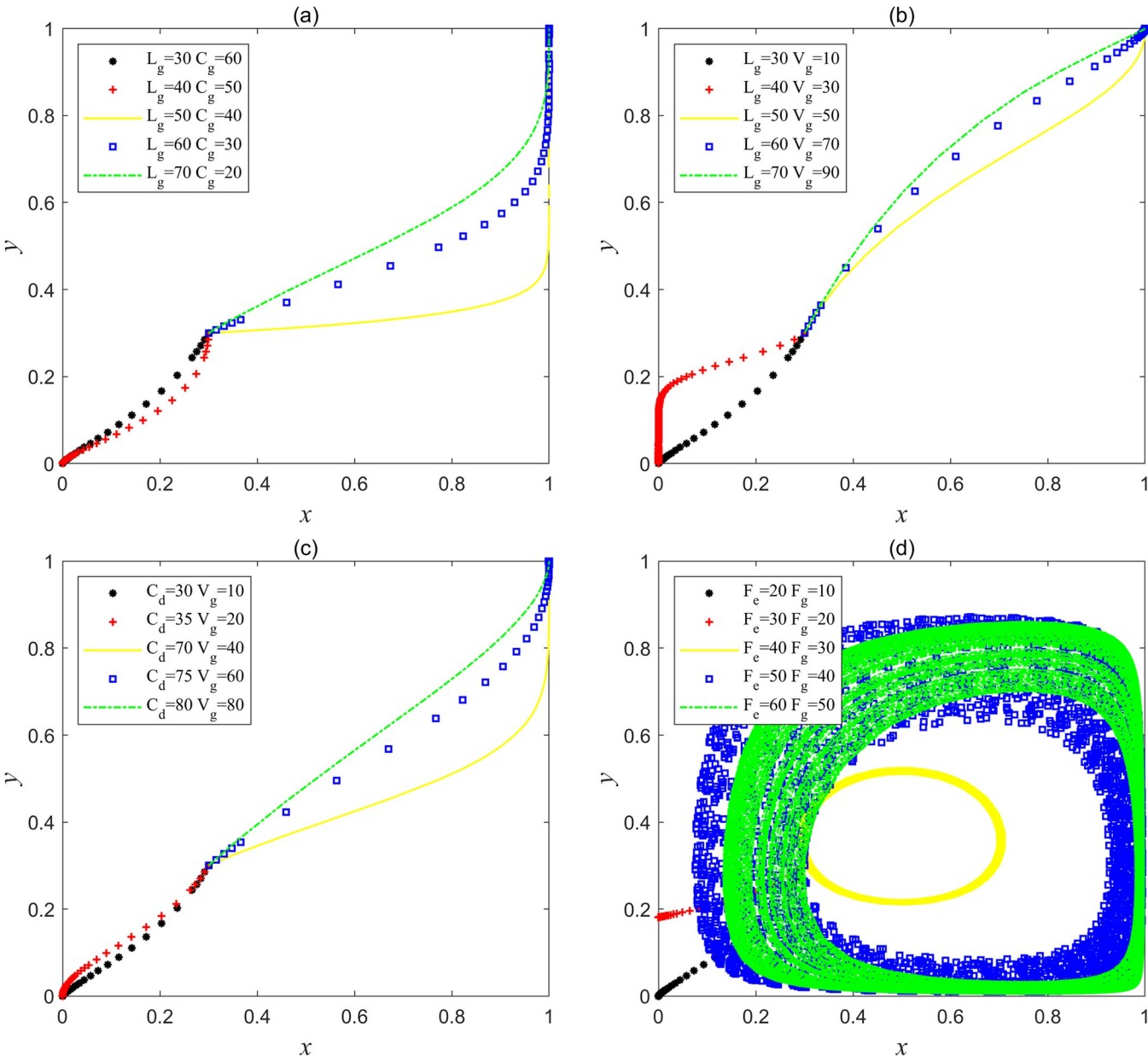

**Fig 5.** Variables adjustment in Scenario 1, where (a) the evolution trajectory when $L_g$ and $C_g$ are both variables; (b) the evolutionary trajectory when $L_g$ and $V_g$ are both variables; (c) the evolutionary trajectory when $C_d$ and $V_g$ are both variables; (d) the evolution trajectory when $F_e$ and $F_g$ are both variables.

time, with the decrease of government regulatory costs or the increase of regulatory benefits, system $I$ will continue to evolve towards optimal equilibrium point, and when $V_g > 50$ and $C_g < 40$, or when $L_g > 50$ and $V_g > 50$, or when $C_d > 70$ and $V_g > 50$, the system $I$ will begin to enter the bad "locked" state from the "invalid" state, and finally tend to the "ideal" equilibrium point (1, 1) state. Fig 5(D) shows that with the increase of the central government's punishment on social capital and government, the system $I$ has evolved from a stable point (0, 0) to an unstable state which is similar to the scenario 2 in Section 4.1, so it indicates that if the punishment on both sides of the game is enhanced, the system $I$ will not tend to optimal evolution, and the punishment function will fail in this scenario.

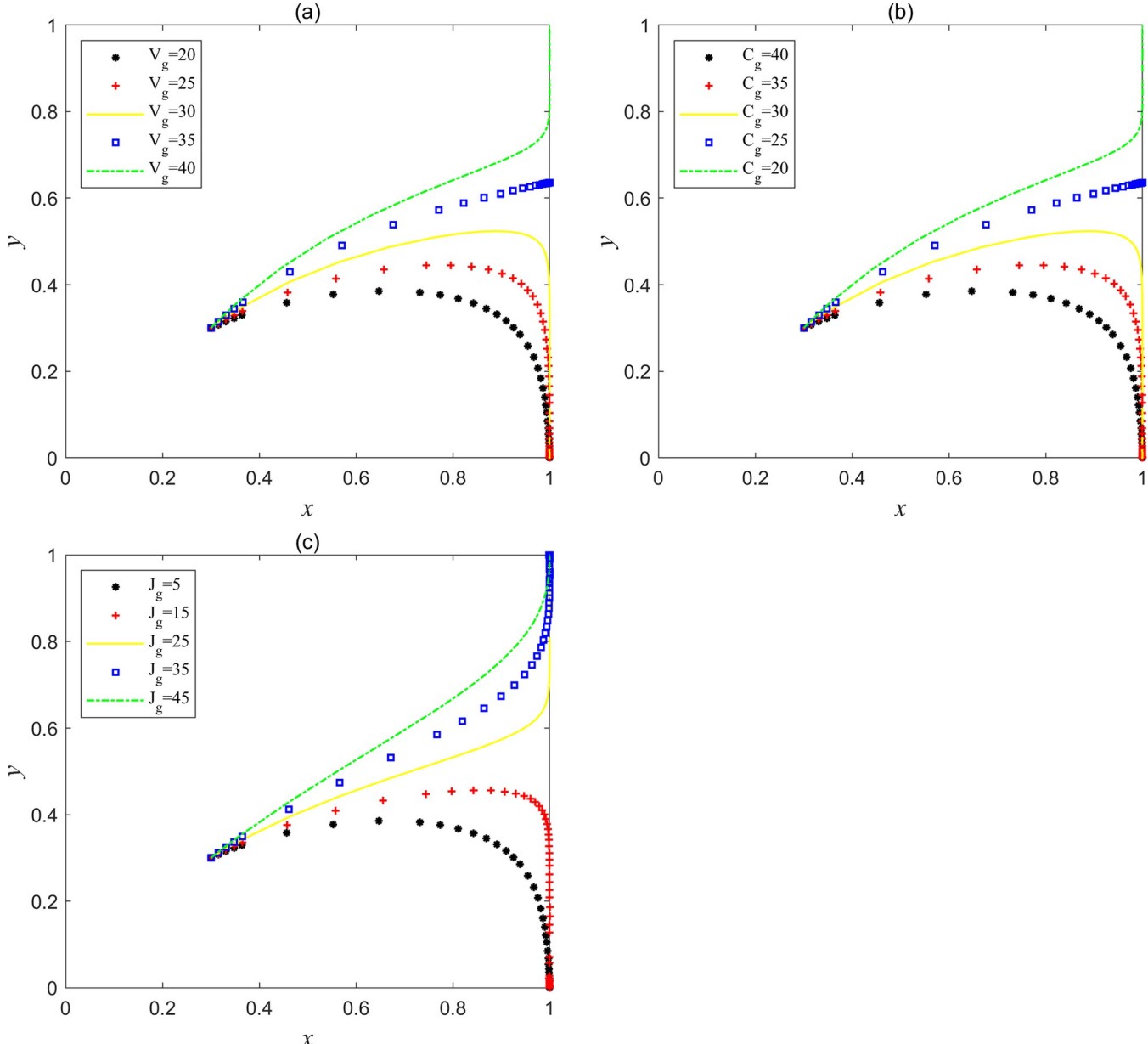

**Fig 6.** Variables adjustment in Scenario 3, where (a) the evolution trajectory when $V_g$ is a variable; (b) the evolutionary trajectory when $C_g$ is a variable; (3) the evolutionary trajectory when $J_g$ is a variable.

**(2) Variable adjustment in Scenario 3.** Based on the parameter variables in Scenario 3, $V_g$, $C_g$ and $J_g$ are set as three groups of variables, and these three groups of variables are simulated by Matlab respectively. The results are shown in Fig 6, when $V_g > 35$ or $Cg < 25$ or $J_g > 25$, system $I$ began to evolve from point $(1, 0)$ to point $(1, 1)$, which indicated that with the increase of government regulatory revenue, the decrease of government regulatory cost or the increase of central government's incentives to local governments, government regulatory authorities began to evolve from "non-supervision " to "supervision", system $I$ can thus jump out of the bad "locked" state and reach the "ideal" stable state point $(1, 1)$.

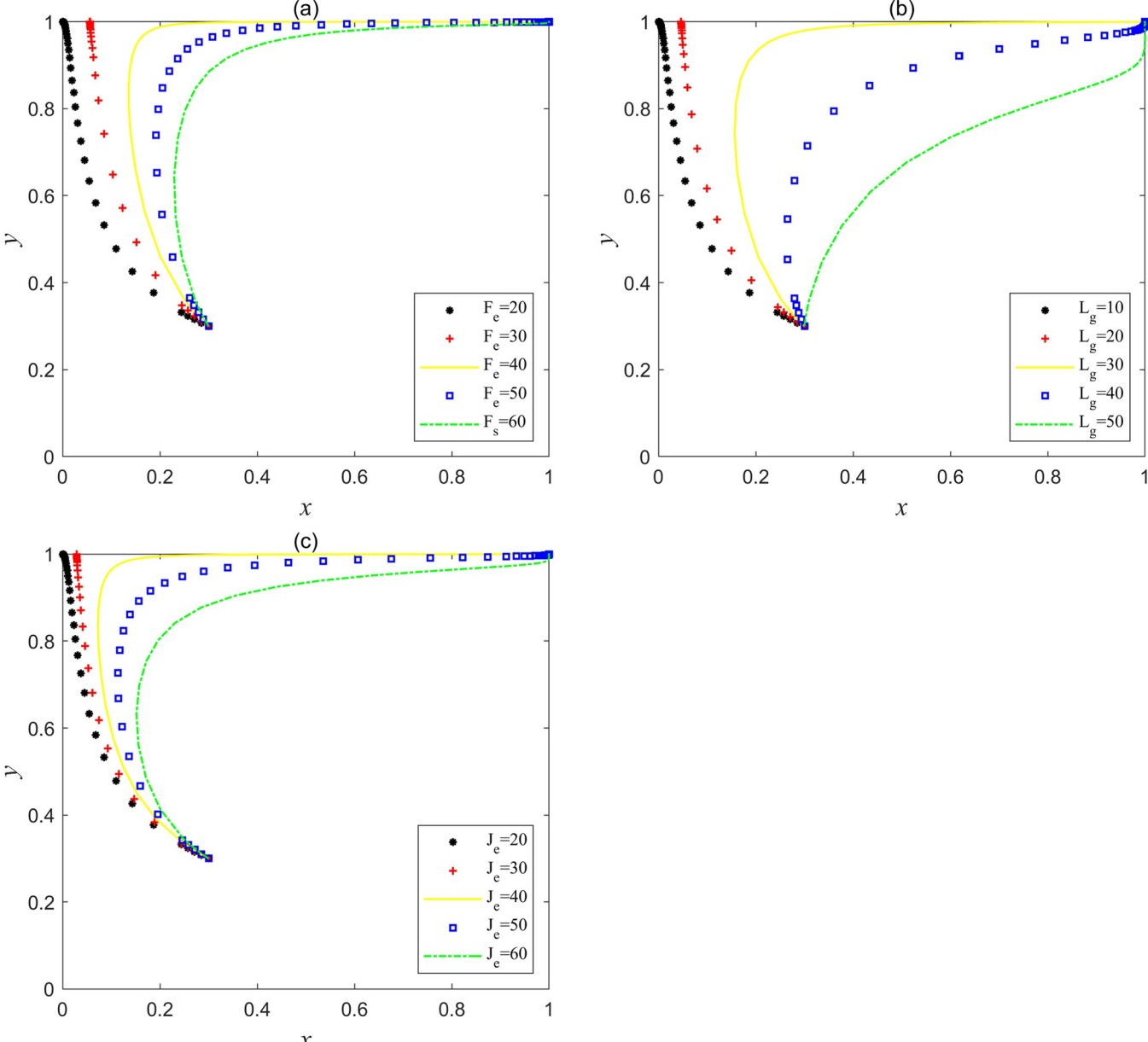

**Fig 7.** Variables adjustment in Scenario 4, where (a) the evolution trajectory when $F_e$ is a variable; (b) the evolutionary trajectory when $L_g$ is a variable; (c) the evolutionary trajectory when $J_e$ is a variable.

**(3) Variable adjustment in Scenario 4.** Based on the parameter variables in Scenario 4, $F_e$, $L_g$ and $J_e$ are set as three groups of variables, and the three groups of variables are simulated by Matlab respectively. The results are shown in Fig 7, when $F_e > 40$, $L_g > 30$ or $J_e > 40$, system $I$ began to evolve from point (0, 1) to point (1, 1). This shows that under the supervision of the government, with the increase of punishment for social capital violations or the increase of operating subsidies for social capital or the increase of incentives for social capital by the central government, social capital will evolve from the strategy "bad service" to the strategy "good service", and system $I$ can also jump out of the bad "lock-in" state and reach the "ideal" stable state point (1, 1).

## 6, Conclusions and suggestions

In this paper, the evolutionary game behavior of two core stakeholders, social capital and government in the PPP model is analyzed, and the evolutionary game equilibrium and simulation analysis of government and social capital under different variables disturbance are discussed. The results show that the mine ecological restoration needs the support of the central government, when the government punishes the low-quality service of social capital lightly, the central government needs to pay higher costs to promote all parties to actively participate in the supervision and operation of the PPP project. It can promote the service quality level of social capital in mine ecological restoration that improvement of government subsidies for "good service" operation of social capital, the increase of penalty cost for breach of operation contract and "bad service" of social capital. The increase of government supervision income and the decrease of supervision cost are helpful to improve the enthusiasm of government for supervision. In the scenario of government supervision, it can effectively spur social capital to improve service quality with increasing the punishment for social capital's breach of contract. However, in the scenario of government non-supervision, the punishment will be ineffective for both illegal social capital and dereliction of duty.

Based on the above conclusions, this paper puts forward the following suggestions:

1. Explore the long-term mechanism and revenue mechanism of mine ecological restoration. According to the results of evolutionary game, it is the key factors to the evolution into an ideal equilibrium state to improve the subsidies and support for social capital to participate in the abandoned mines ecological restoration, which are the long-term mechanism and income mechanism for social capital to actively participate in the ecological restoration of mines. For example, the central government gives social capital extra operating income by giving it the right to use natural resources assets for a certain period of time, and to enrich publicity channels and strengthen intangible incentives such as reputation and word of mouth for social capital.

2. Restrain the behavior of social capital and raise the cost of social capital violation. According to the results of evolutionary game, if the punishment is not obvious, the possible benefits of illegal behavior will induce social capital to take risk-taking strategies. With increasing the punishment of local governments for illegal behavior of social capital, it will promote social capital to actively provide high-quality services. So that government can not only obtain social benefits and green environmental effects, but also get higher central government rewards. At the same time, the increase in income obtained through the punishment of violations can offset the government's some regulatory costs, which also improve the enthusiasm of supervision and ensure sufficient incentive for supervision.

3. Absorb the third party to improve the supervision procedure, reduce the supervision cost and improve the supervision efficiency. Government can actively introduce the public and other third-party forces to participate in the PPP project of the abandoned mines and surrounding wasteland, such as, improving government information disclosure, smoothing reporting channels and forming a public and media supervision and disclosure mechanism. At the same time, the existence of third-party forces, such as the public, will play a reverse supervision mechanism on the government's own behavior, and prevent the government from taking negative actions or inaction.

4. Strengthen the communication between the two sides of the PPP project. On the one hand, it can reduce the difficulty of government supervision, on the other hand, it can encourage the government to know and recognize when the implementation and operation of mine

ecological restoration are difficult, which can reduce the government's unnecessary punishment for social capital, so that social capital can get more help from the central government when they choose to provide high-quality services.

In addition, the quality supervision of PPP projects related to mining environmental governance in China is a game process involving multiple stakeholders, and the study only considers two entities: government regulatory agencies and the private sector. Subsequent research will include third-party evaluation institutions (referred to as "third parties") and residents as game subjects, and construct a dynamic evolutionary game model of "government private sector residents" or "government third party private sector". More factors will also be considered, such as analyzing "public participation" and "reputation incentives" as variables in the model, Explore the inherent mechanism of the multi-dimensional quality supervision model for PPP projects related to mining environmental governance.

## Supporting information

**S1 Appendix.**
(DOCX)

**S2 Appendix. Parameters of scenarios.**
(XLSX)

## Author Contributions

**Conceptualization:** Dongmei Feng.

**Data curation:** Liang Wang.

**Methodology:** Liang Wang.

**Software:** Liang Wang, Xiumei Duan.

**Supervision:** Dongmei Feng.

**Validation:** Xiumei Duan.

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
