## [Decision Letter · Decision Letter 0]

26 Dec 2023

PONE-D-23-30222Game analysis on PPP model operation of abandoned mines ecological restoration under the innovation of central government's reward and punishment system in ChinaPLOS ONE

Dear Dr. Wang,

Thank you for submitting your manuscript to PLOS ONE. After careful consideration, we feel that it has merit but does not fully meet PLOS ONE’s publication criteria as it currently stands. Therefore, we invite you to submit a revised version of the manuscript that addresses the points raised during the review process.

We look forward to receiving your revised manuscript.

Kind regards,

Xingwei Li, Ph.D.

Academic Editor

PLOS ONE

4. We note that your Data Availability Statement is currently as follows: [ll relevant data are within the manuscript and its Supporting Information files.]

Reviewers' comments:

Reviewer's Responses to Questions

**Comments to the Author**

1. Is the manuscript technically sound, and do the data support the conclusions?

Reviewer #1: Yes

Reviewer #2: Partly

Reviewer #3: Yes

2. Has the statistical analysis been performed appropriately and rigorously? 

Reviewer #1: Yes

Reviewer #2: No

Reviewer #3: N/A

3. Have the authors made all data underlying the findings in their manuscript fully available?

Reviewer #1: Yes

Reviewer #2: No

Reviewer #3: Yes

4. Is the manuscript presented in an intelligible fashion and written in standard English?

Reviewer #1: Yes

Reviewer #2: No

Reviewer #3: Yes

5. Review Comments to the Author

Reviewer #1: The author did a very good job.

1.The theme of the article is innovative.

2.The manuscript technically sound, and the data support the conclusions.

3.Statistical analysis appropriately and rigorously.

4.The article is easy to understand and has no grammar errors.

Reviewer #2: In this paper, an evolutionary game model of "local government-social capital" was developed to analyze the interaction and behavior mechanism of core stakeholders in the operation process of abandoned mines ecological restoration PPP mode. It has some significance. However, J have the following concerns

1) --Literature Review should be presented as a separated section. Essentially, many papers have examined the evolutionary game .

For example：

--Zhang S., Wang, C.X., Yu C., The evolutionary game analysis and simulation with system dynamics of manufacturer's emissions abatement behavior under cap-and-trade regulation, Applied Mathematics and Computation, 2019, 355, 343-355.

--Shi Y., Wei Z., Shahbaz M., Zeng Y., Exploring the dynamics of low-carbon technology diffusion among enterprises: An evolutionary game model on a two-level heterogeneous social network, 2021,101: 105399.

--Jiang N., Feng Y., Wang X., Fractional-order evolutionary game of green and low-carbon innovation in manufacturing enterprises, Alexandria Engineering Journal, 2022,61(2), 12673-12687.

It is suggested that author(s) introduce the papers related to the above evolutionary game analysis into the literature review, and identify the differences between the previous papers and current paper. Thus, the authors have to differentiate further and rethink about their work's innovations.

2) The innovation points for this paper should be clearly given in Introduction.

3) In Section 4 Numerical simulation analysis , where the initial values of the parameter comes? Explanations of the basis for setting the values of these parameters should be presented.

4) In Section 5. Conclusions and suggestions， the future research directions should be presented.

Reviewer #3: The authors research the interaction and behavior mechanism of core stakeholders (social capital, local government and central government) in the ecological restoration of abandoned mine from the perspective of the PPP model. The hypotheses are reasonable, the proofs are relatively complete and the results are clear.

This article is mainly a simulation analysis and has little to do with the statistical analysis, so I choose “N/A” in Review Question 2.

But there are six issues that the authors need to revise.

1. The narrative order about the literature review needs to be adjusted.

The authors focus on the ecological restoration of abandoned mine from the perspective of an economic approach, so after providing the background information, the authors should review the traditional perspectives and approaches in the previous researches on the ecological restoration of abandoned mine and then put forward the PPP model given that this topic lacks an economic perspective.

The brief review of the PPP model and the researches with it should be presented after putting forward the PPP model.

When finishing the above work, the authors need to elaborate “However, the ecological environment restoration of mines generally lacks a good income mechanism, and social capital is still in the wait-and-see and exploration stage at present”. Based on this work, “it is of great research value to explore the application of PPP model innovation in low-profit mine ecological restoration projects” can be better understood.

So how to adjust the Introduction is clear. The first and last paragraph need not to be adjusted. The middle content can be conducted as follows: (1) Review the traditional approaches and put forward the PPP model; (2) Brief review the PPP model and the researches with it; (3) Elaborate the research gap.

2. The shortcomings (and contributions if they are important) in the relevant literature should be clear instead of “someone did something, some did something…”.

The authors can criticize the previous researches from method or evidence to highlight the research value.

3. The hypotheses should be concise. The authors can explain the logic of a hypothesis before presenting the hypothesis. Do not put the hypothesis and its explanation together.

4. Why choose the evolutionary game model to explain the operation of PPP projects? or how the operation of PPP projects manifests the characteristics of evolutionary games? The authors need to expound the connection between the evolutionary game model and the operation of PPP projects, or summarize several stylized facts of PPP projects, which are in accord with the characteristics of evolutionary games. This work can be put in the Introduction or Model hypothesis and construction.

5. Given that the PPP model has been conducted in reality, how do the authors interpret if the actual operation does not reach to “equilibrium”? Consider giving some possible reasons.

6. The proofs need to be presented in details. The authors can put the detailed process in the appendix. Otherwise, we can hardly understand the game result without patience and enthusiasm.

6. PLOS authors have the option to publish the peer review history of their article (what does this mean?). If published, this will include your full peer review and any attached files.

Reviewer #1: No

Reviewer #2: No

Reviewer #3: No

---

## [Author Response · Author response to Decision Letter 0]

12 Feb 2024

To Reviewer 1:

Thank you to this reviewer for their summary and recognition of our work, as well as for their efforts and positive feedback during the review of our paper.

To Reviewer 2:

Opinion 1: In this paper, an evolutionary game model of "local government-social capital" was developed to analyze the interaction and behavior mechanism of core stakeholders in the operation process of abandoned mines ecological restoration PPP mode. It has some significance. However, J have the following concerns

1) --Literature Review should be presented as a separated section. Essentially, many papers have examined the evolutionary game .

For example：

--Zhang S., Wang, C.X., Yu C., The evolutionary game analysis and simulation with system dynamics of manufacturer's emissions abatement behavior under cap-and-trade regulation, Applied Mathematics and Computation, 2019, 355, 343-355.

--Shi Y., Wei Z., Shahbaz M., Zeng Y., Exploring the dynamics of low-carbon technology diffusion among enterprises: An evolutionary game model on a two-level heterogeneous social network, 2021,101: 105399.

--Jiang N., Feng Y., Wang X., Fractional-order evolutionary game of green and low-carbon innovation in manufacturing enterprises, Alexandria Engineering Journal, 2022,61(2), 12673-12687.

It is suggested that author(s) introduce the papers related to the above evolutionary game analysis into the literature review, and identify the differences between the previous papers and current paper. Thus, the authors have to differentiate further and rethink about their work's innovations.

Argue and Revise: Thank you for pointing this out. Regarding the review section in the introduction of the paper, the authors have undertaken a substantial revision. In accordance with the recommendations of the reviewers, the review section has been crafted into an independent chapter. Furthermore, based on the detailed suggestions of another reviewer, adjustments have been made to the narrative logic, as outlined in LOA2. The revised structure involves (1) a review of traditional approaches leading to the proposal of the PPP model, (2) a concise overview of the PPP model and related studies, and finally, (3) an exposition of research gaps and the introduction of the innovative aspects of our paper. In the section reviewing the PPP model, we have consulted and cited literature provided by the reviewers. We have elaborated on the latest developments in the integration of evolutionary game models across various domains, drawing inspiration for our paper. Additionally, we have argued evolutionary game models to address the issue of interest distribution among multiple stakeholders in PPP relationships, delving into the specific realm of environmental governance in mining projects. We have discussed the progress and challenges in this field, incorporating insights from the literature review and concluding with the presentation of our research's innovative contributions and distinctive features. Please refer to LOA2 in the revised manuscript for a detailed account of these modifications.

Opinion 2: The innovation points for this paper should be clearly given in Introduction.

Revise: Thank you for pointing this out. The authors have undertaken a revision of this section, and in the final part of the revised manuscript's second chapter - the literature review, they have provided an explanation of the innovative aspects and distinctive features of their research. Please refer to the modified content in that section for further details, see LOA3.

Opinion 3: In Section 4 Numerical simulation analysis , where the initial values of the parameter comes? Explanations of the basis for setting the values of these parameters should be presented.

Argue and Reply: Thank you for pointing this out. The study investigates the strategic interactions and adjustments between government regulatory agencies and private sectors, employing Matlab simulation to analyze the evolutionary impact of equilibrium strategies and model parameter adjustments on the strategic choices of both parties in different scenarios. The aim of the paper is to abstract key variables for analyzing the various equilibrium relationships formed from the game dynamics between the two parties in PPP projects. This provides a more intuitive understanding of the dynamic evolution trends in the quality supervision process of ecological restoration in mining under PPP operations. Based on the conclusions from the analysis, the paper suggests measures to enhance the service quality of the PPP operational model for mining ecological restoration from the perspective of government regulation. The focus of this article is primarily on simulation analysis and has limited connection with statistical analysis from actual cases. In the revised manuscript's Section 5(Section 4 in origin manuscript), which meets the parameter requirements of Assumption 2 in Section 4, different scenarios for the operational model of mining ecological restoration PPP can be assigned based on project-specific conditions, and the values chosen do not represent the actual costs or benefits for government regulatory agencies and private capital in mining ecological restoration PPP projects. If there are actual cases of PPP projects for mining environmental restoration with operational data, the approach presented in this paper can be utilized to validate and retrospectively analyze real projects from the perspective of game theory, shedding light on the reasons for project success or failure.

Opinion 4: In Section 5. Conclusions and suggestions，the future research directions should be presented.

Revise: Thank you for pointing this out. The authors have supplemented this section based on the suggestions of the reviewers. Please refer to the final section of Chapter 6 of the revised manuscript, see LOA8.

To Reviewer 3:

Opinion 1: The narrative order about the literature review needs to be adjusted. The authors focus on the ecological restoration of abandoned mine from the perspective of an economic approach, so after providing the background information, the authors should review the traditional perspectives and approaches in the previous researches on the ecological restoration of abandoned mine and then put forward the PPP model given that this topic lacks an economic perspective. The brief review of the PPP model and the researches with it should be presented after putting forward the PPP model. When finishing the above work, the authors need to elaborate “However, the ecological environment restoration of mines generally lacks a good income mechanism, and social capital is still in the wait-and-see and exploration stage at present”. Based on this work, “it is of great research value to explore the application of PPP model innovation in low-profit mine ecological restoration projects” can be better understood. So how to adjust the Introduction is clear. The first and last paragraph need not to be adjusted. The middle content can be conducted as follows: (1) Review the traditional approaches and put forward the PPP model; (2) Brief review the PPP model and the researches with it; (3) Elaborate the research gap.

Revise and Reply: Thank you for pointing this out. Regarding the review section of the paper, the authors have rewritten it. According to the reviewer's suggestions, adjustments have been made to the narrative logic in this revised section of the paper. And based on the suggestions of another reviewer, this section separates the overall cutting-edge content into two separate sections, namely the introduction and literature review. The details of the modifications can be found in the description of this section of the revised manuscript, see LOA1.

Opinion 2: The shortcomings (and contributions if they are important) in the relevant literature should be clear instead of “someone did something, some did something…”.

The authors can criticize the previous researches from method or evidence to highlight the research value.

Revise and Reply: Thank you for pointing this out. The content of this section has been rewritten, and based on the suggestions of the reviewers, the narrative logic has been modified and comments on previous literature have been added. For more details, please refer to the modifications made to this section in the revised manuscript. The citations of previous literature are reflected in the narrative logic of each section. For example, in the review of the PPP model section, we read and cited the literature materials provided by the reviewers, and elaborated on and explained the latest research situation of the combination of evolutionary game models in multi domain practice, in order to provide us with inspiration points and innovative possibilities for transplanting to a new field in our paper. In addition, The application of evolutionary game models in the study of the distribution of interests among multiple stakeholders involved in public-private partnerships such as PPP is explored. This paper delves into the special field of mining environmental governance projects, as well as the progress and existing problems in this field of research. At the end of the literature review, innovative points and characteristics of my own research are proposed, as shown in LOA1.

Opinion 3: The hypotheses should be concise. The authors can explain the logic of a hypothesis before presenting the hypothesis. Do not put the hypothesis and its explanation together.

Revise and Reply: Thank you for pointing this out. We have made modifications to this section based on the comments of the reviewers. Please refer to Section 3 of the revised manuscript for details, see LOA4.

Opinion 4: Why choose the evolutionary game model to explain the operation of PPP projects? or how the operation of PPP projects manifests the characteristics of evolutionary games? The authors need to expound the connection between the evolutionary game model and the operation of PPP projects, or summarize several stylized facts of PPP projects, which are in accord with the characteristics of evolutionary games. This work can be put in the Introduction or Model hypothesis and construction.

Argue and Revise: Thank you for pointing this out. The authors have seen that in the operation of the PPP project for ecological restoration of abandoned mines, whether from the perspective of risk sharing and profit distribution, or from the regulatory perspective, there are game relationships among all parties to ensure the smooth progress of the PPP project. Starting from this idea, the authors gradually rationalize the operating mechanism of PPP in this field, analyze the contractual relationship and project participants in the PPP project for ecological restoration of abandoned mines, and construct a game framework for both government public sectors and social capital investors. The authors have rewritten this section based on the comments of the reviewers, detailing the content of Section 1 and 2 of the revised manuscript. The authors have reflected these narrative logics in the revised manuscript, for example, in the review of the PPP model section, explaining the latest research on the combination of evolutionary game models in multi-domain practice, as well as the various applications of evolutionary game models in PPP projects. Finally, they delve into the special field of mining environmental governance projects, as well as the progress and existing problems in this field of research. Finally, they propose their own innovative points and characteristics in the literature review, see LOA2.

Opinion 5: Given that the PPP model has been conducted in reality, how do the authors interpret if the actual operation does not reach to “equilibrium”? Consider giving some possible reasons.

Argue and Reply: Thank you for pointing this out. The dynamic evolutionary game in the paper is a benefit matrix obtained by assuming conditions, and finally a replicated dynamic equation is obtained. Through the analysis of equilibrium points and evolutionary strategy stability, 5 equilibrium points of the system are obtained. These 5 equilibrium points are analyzed and proven in sections 4.1 and 4.2. Through a detailed analysis of these equilibrium points, it can be concluded that different project parameters are more favorable to the government in some cases and more favorable to the social capital in others. From the basic game logic, we hope that this game equilibrium can achieve the optimal effect, that is, both parties can profit the most. Therefore, we have redefined some more optimized combination conditions to meet this goal. In addition, Section 5 of the revised manuscript also mentions "Scenario 2", in which the system does not have an evolutionary stable strategy, that is, it cannot reach an evolutionary equilibrium strategy. According to Friedman's theory, the stability of Scenario 2 is analyzed in Table 3, and the reason for this instability is explained through numerical simulation in Section 5.1. For detailed explanation, please refer to "Scenario 2" in Section 5.1. Under the condition of Scenario 2, the behavior evolution of government regulatory agencies and social capital is in a cyclical swing state, and will not form or reach a stable evolution strategy, that is, cannot reach an equilibrium state between the two. This also explains some states from a practical perspective: during the high incidence period of ecological restoration service quality issues in mines in China, most government regulatory departments adopt an active and active regulatory model in pursuit of maximizing social benefits. However, during the low incidence period of accidents, in order to ensure the interests of their own departments, the extremely intermittent regulatory model is abolished, The central government's commitment to reward local governments and social capital cannot serve as a stable strategic choice for both sides, similar to a game of cat and mouse turning trap.

Opinion 6: The proofs need to be presented in details. The authors can put the detailed process in the appendix. Otherwise, we can hardly understand the game result without patience and enthusiasm.

Revise and Reply: Thank you for pointing this out. The authors have supplemented this section based on the suggestions of the reviewers. Please refer to the appendix of the revised manuscript. And supplemented Table 3 of the original manuscript. We have also revised some of the issues in the original manuscript and made some adjustments in the later calculation section. Please refer to Section 5 of the revised manuscript for details, see LOA5, LOA6 and LOA7.

---

## [Decision Letter · Decision Letter 1]

30 Apr 2024

PONE-D-23-30222R1Game analysis on PPP model operation of abandoned mines ecological restoration under the innovation of central government's reward and punishment system in ChinaPLOS ONE

Dear Dr. Wang,

Thank you for submitting your manuscript to PLOS ONE. After careful consideration, we feel that it has merit but does not fully meet PLOS ONE’s publication criteria as it currently stands. Therefore, we invite you to submit a revised version of the manuscript that addresses the points raised during the review process.

We look forward to receiving your revised manuscript.

Kind regards,

Xingwei Li, Ph.D.

Academic Editor

PLOS ONE

Additional Editor Comments:

Reconsider the "Option 4" and "Argue and Revise". What are the possible reasons if the actual operation does not reach to “equilibrium”? Given that China has a large number of administrative units, you can analyze why some administrative units adopt a kind of regulation and why the others administrative units do not adopt this kind of regulation during the different incidence period. The authors' response, "during the high incidence period of ecological restoration service quality issues in mines in China, most government regulatory departments adopt an active and active regulatory model in pursuit of maximizing social benefits. However, during the low incidence period of accidents, in order to ensure the interests of their own departments, the extremely intermittent regulatory model is abolished", just gives the theoretical reasons or the theoretical assumptions, ignores the reality explanations. To make the model in this paper more applied, the authors can give some stylized facts or cases that clearly manifest the actions of the governments.

Reviewers' comments:

Reviewer's Responses to Questions

**Comments to the Author**

1. If the authors have adequately addressed your comments raised in a previous round of review and you feel that this manuscript is now acceptable for publication, you may indicate that here to bypass the “Comments to the Author” section, enter your conflict of interest statement in the “Confidential to Editor” section, and submit your "Accept" recommendation.

Reviewer #3: (No Response)

2. Is the manuscript technically sound, and do the data support the conclusions?

Reviewer #3: Yes

3. Has the statistical analysis been performed appropriately and rigorously? 

Reviewer #3: N/A

4. Have the authors made all data underlying the findings in their manuscript fully available?

Reviewer #3: Yes

5. Is the manuscript presented in an intelligible fashion and written in standard English?

Reviewer #3: Yes

6. Review Comments to the Author

Reviewer #3: Reconsider the "Option 4" and "Argue and Revise". What are the possible reasons if the actual operation does not reach to “equilibrium”? Given that China has a large number of administrative units, you can analyze why some administrative units adopt a kind of regulation and why the others administrative units do not adopt this kind of regulation during the different incidence period. The authors' response, "during the high incidence period of ecological restoration service quality issues in mines in China, most government regulatory departments adopt an active and active regulatory model in pursuit of maximizing social benefits. However, during the low incidence period of accidents, in order to ensure the interests of their own departments, the extremely intermittent regulatory model is abolished", just gives the theoretical reasons or the theoretical assumptions, ignores the reality explanations. To make the model in this paper more applied, the authors can give some stylized facts or cases that clearly manifest the actions of the governments.

Whether to modify this issue depends on the author's wishes.

7. PLOS authors have the option to publish the peer review history of their article (what does this mean?). If published, this will include your full peer review and any attached files.

Reviewer #3: No

---

## [Author Response · Author response to Decision Letter 1]

7 May 2024

To Reviewer 3:

Reviewer #3: Reconsider the "Option 4" and "Argue and Revise". What are the possible reasons if the actual operation does not reach to “equilibrium”? Given that China has a large number of administrative units, you can analyze why some administrative units adopt a kind of regulation and why the others administrative units do not adopt this kind of regulation during the different incidence period. The authors' response, "during the high incidence period of ecological restoration service quality issues in mines in China, most government regulatory departments adopt an active and active regulatory model in pursuit of maximizing social benefits. However, during the low incidence period of accidents, in order to ensure the interests of their own departments, the extremely intermittent regulatory model is abolished", just gives the theoretical reasons or the theoretical assumptions, ignores the reality explanations. To make the model in this paper more applied, the authors can give some stylized facts or cases that clearly manifest the actions of the governments.

Whether to modify this issue depends on the author's wishes.

Argue and Reply: Thank you for pointing this out. The authors apologize that their response to this question may not have been clear and thorough enough before. The previous response to this question provided a theoretical and detailed explanation. Please refer to the previous detailed response to this question, and it also provided an explanation for a typical programmatic failure situation of the government in the supervision of many PPP projects in reality, that is, most governments adopt a movement style supervision mode in pursuit of maximizing social benefits, which is consistent with the description of the model's Scenario 2 results, and this is also a typical failure situation in many PPP project supervision cases. 

The authors provide the following explanations and revisions to the questions raised by the reviewer:

Firstly, reply to the question: Given that China has a large number of administrative units, you can analyze why some administrative units adopt a kind of regulation and why the others administrative units do not adopt this kind of regulation during the different incidence period.

The two objects studied in the paper (government departments and social capitals) are both groups that change over time. The factors that affect the changes of these two groups have both regularity presented through the selection process during evolution (the selection process has a certain inertia), and at the same time, this process also harbors the driving force of mutation. Mutation is a learning and imitation process that constantly trial and error. When all participating entities in the game between government departments and social capitals adopt the same behavior strategy respectively, such as when the behavior strategy of certain entities in the government department group undergoes a sudden change (such as when facing "good service" provided by certain social capital, the government department chooses to negative supervision), if the benefits brought by the behavior strategy of that entity are higher than those of other entities in the group, then other participating government department entities will also follow the behavior strategy of the mutated entity and adopt similar behavior strategies. As a result, the proportion of individuals adopting the new strategy in the entire game group will increase. As the game progresses and time passes, the behavior strategy with higher returns will gradually replace the behavior strategy with lower returns, the individual experience of government departments or social capital adopting this strategy is increasing, but there will be some inertia in the game process that makes the evolutionary game process impossible to achieve instantaneously. That is, it will not happen immediately. All government departments or social capital will adopt the same strategy at the same time, and there is a process of change. Therefore, in different periods of occurrence, some administrative units will adopt certain regulations, while other administrative units will not adopt such regulations. In addition, Section 5.1 of the paper also mentions "Scenario 2", in which the system does not have an evolutionarily stable strategy, that is, it is impossible to achieve an equilibrium strategy of evolution. Under the conditions of "Scenario 2", the behavior evolution of government departments and social capitals are in a cyclical swing state, and it will not form or reach a stable evolution strategy, that is, it will not reach an equilibrium state between the two groups, and a cyclical game strategy cycle will be formed between the two groups.

Secondly, reply to the question: just gives the theoretical reasons or the theoretical assumptions, ignores the reality explanations. To make the model in this paper more applied, the authors can give some stylized facts or cases that clearly manifest the actions of the governments.

In actual PPP project implementation, project implementation failures often occur, and a typical stylized case is the game of "cat and mouse" that often occurs in government departments and social capitals in supervision. In China, during the high-incidence period of ecological restoration service quality problems in mines, most government departments adopt an active regulatory model in pursuit of maximizing social benefits. In the low-incidence period of accidents, in order to ensure the interests of their own departments, negative regulatory models are adopted. The authors have made modifications to this part and correspondingly cited some stylized facts or case analysis results as evidence. The model established in this article simulates Scenario 2, which provides a good explanation for this imbalance in reality. Detailed modifications can be found in LOA1.

---

## [Editor Report · Decision Letter 2]

13 May 2024

Game analysis on PPP model operation of abandoned mines ecological restoration under the innovation of central government's reward and punishment system in China

PONE-D-23-30222R2

Dear Dr. Wang,

We’re pleased to inform you that your manuscript has been judged scientifically suitable for publication and will be formally accepted for publication once it meets all outstanding technical requirements.

Kind regards,

Xingwei Li, Ph.D.

Academic Editor

PLOS ONE
---

## [Editor Report · Acceptance letter]

16 May 2024

PONE-D-23-30222R2 

PLOS ONE

Dear Dr. Wang, 

I'm pleased to inform you that your manuscript has been deemed suitable for publication in PLOS ONE. Congratulations! Your manuscript is now being handed over to our production team.

Kind regards, 

on behalf of

Prof. Dr. Xingwei Li 

Academic Editor

PLOS ONE